# NCBI’s Virus Discovery Hackathon: Engaging Research Communities to Identify Cloud Infrastructure Requirements

**DOI:** 10.3390/genes10090714

**Published:** 2019-09-16

**Authors:** Ryan Connor, Rodney Brister, Jan P. Buchmann, Ward Deboutte, Rob Edwards, Joan Martí-Carreras, Mike Tisza, Vadim Zalunin, Juan Andrade-Martínez, Adrian Cantu, Michael D’Amour, Alexandre Efremov, Lydia Fleischmann, Laura Forero-Junco, Sanzhima Garmaeva, Melissa Giluso, Cody Glickman, Margaret Henderson, Benjamin Kellman, David Kristensen, Carl Leubsdorf, Kyle Levi, Shane Levi, Suman Pakala, Vikas Peddu, Alise Ponsero, Eldred Ribeiro, Farrah Roy, Lindsay Rutter, Surya Saha, Migun Shakya, Ryan Shean, Matthew Miller, Benjamin Tully, Christopher Turkington, Ken Youens-Clark, Bert Vanmechelen, Ben Busby

**Affiliations:** 1National Center for Biotechnology Information, National Library of Medicine, National Institutes of Health, Bethesda MD 20894, USA; connorrp@ncbi.nlm.nih.gov (R.C.); jamesbr@ncbi.nlm.nih.gov (R.B.); zaluninvv@ncbi.nlm.nih.gov (V.Z.); efremova2@ncbi.nlm.nih.gov (A.E.); fleischmannlc@ncbi.nlm.nih.gov (L.F.); leubsdor@ncbi.nlm.nih.gov (C.L.); 2Charles Perkins Centre, School of Life and Environmental Sciences, The University of Sydney, Sydney, NSW 2006, Australia; jan.buchmann@sydney.edu.au; 3KU Leuven, Department of Microbiology & Immunology, Rega Institute, Leuven BE3000, Belgium; ward.deboutte@kuleuven.be (W.D.); joan.marti@kuleuven.be (J.M.-C.); bert.vanmechelen@kuleuven.be (B.V.); 4Department of Biology, San Diego State University, 5500 Campanile Dr., San Diego, CA 92182, USA; redwards@sdsu.edu (R.E.); vcantualessioroble@sdsu.edu (A.C.); mgiluso@sdsu.edu (M.G.); margaret.henderson@sdsu.edu (M.H.); klevi@sdsu.edu (K.L.); slevi3509@sdsu.edu (S.L.); 5Lab of Cellular Oncology, NCI, NIH, Bethesda, MD 20892-4263, USA; mike.tisza@nih.gov; 6Research Group on Computational Biology and Microbial Ecology, Department of Biological Sciences, Universidad de los Andes, Bogotá 111711, Colombia; js.andrade10@uniandes.edu.co (J.A.-M.); lm.forero10@uniandes.edu.co (L.F.-J.); 7Max Planck Tandem Group in Computational Biology, Universidad de los Andes, Bogotá 111711, Colombia; 8D’Amour & Associates, 11839 Hilltop Drive, Los Altos Hills, CA 94024, USA; mike@damourventures.com; 9Department of Genetics, University Medical Center Groningen, Groningen 9713AV, The Netherlands; s.garmaeva@umcg.nl; 10Computational Bioscience Program, University of Colorado Anschutz, Aurora, CO 80045, USA; cody.glickman@ucdenver.edu; 11Bioinformatics and Systems Biology Program, University of California at San Diego, 9500 Gilman Dr., La Jolla, CA 92093, USA; bkellman@eng.ucsd.edu; 12Department of Biomedical Engineering, University of Iowa, Iowa City, IA 52242, USA; dk1313t63@gmail.com; 13Division of Infectious Diseases, Department of Medicine, Vanderbilt University Medical Center, Nashville, TN 37232, USA; suman.b.pakala@vumc.org; 14Department of Laboratory Medicine, University of Washington Virology, 1616 Eastlake Ave E, Seattle, WA 98102, USA; vpeddu@uw.edu (V.P.); rcs333@uw.edu (R.S.); 15Department of Biosystems Engineering, University of Arizona, Tucson, AZ 85716, USA; aponsero@email.arizona.edu (A.P.); mattmiller899@email.arizona.edu (M.M.); kyclark@email.arizona.edu (K.Y.-C.); 16MITRE Corporation, 7515 Colshire Drive, McLean, VA 22102-7539, USA; eribeiro@mitre.org; 17Department of Biostatistics, Harvard T.H. Chan School of Public Health, Boston, MA 02115, USA; froy@hsph.harvard.edu; 18University of Tsukuba, Ibaraki 305-8575, Japan; lindsayannerutter@gmail.com; 19Boyce Thompson Institute, Ithaca, NY 14853, USA; ss2489@cornell.edu; 20Bioscience Division, Los Alamos National Lab, Los Alamos, NM 87545, USA; migun@lanl.gov; 21Center for Dark Energy Biosphere Investigations, University of Southern California, Los Angeles, CA 90089, USA; tully.bj@gmail.com; 22School of Natural Sciences, University of California Merced, Merced, CA 95343, USA; cturkington@ucmerced.edu

**Keywords:** metagenomic, viruses, SRA, STRIDES, hackathon, infrastructure, cloud computing

## Abstract

A wealth of viral data sits untapped in publicly available metagenomic data sets when it might be extracted to create a usable index for the virological research community. We hypothesized that work of this complexity and scale could be done in a hackathon setting. Ten teams comprised of over 40 participants from six countries, assembled to create a crowd-sourced set of analysis and processing pipelines for a complex biological data set in a three-day event on the San Diego State University campus starting 9 January 2019. Prior to the hackathon, 141,676 metagenomic data sets from the National Center for Biotechnology Information (NCBI) Sequence Read Archive (SRA) were pre-assembled into contiguous assemblies (contigs) by NCBI staff. During the hackathon, a subset consisting of 2953 SRA data sets (approximately 55 million contigs) was selected, which were further filtered for a minimal length of 1 kb. This resulted in 4.2 million (Mio) contigs, which were aligned using BLAST against all known virus genomes, phylogenetically clustered and assigned metadata. Out of the 4.2 Mio contigs, 360,000 contigs were labeled with domains and an additional subset containing 4400 contigs was screened for virus or virus-like genes. The work yielded valuable insights into both SRA data and the cloud infrastructure required to support such efforts, revealing analysis bottlenecks and possible workarounds thereof. Mainly: (i) Conservative assemblies of SRA data improves initial analysis steps; (ii) existing bioinformatic software with weak multithreading/multicore support can be elevated by wrapper scripts to use all cores within a computing node; (iii) redesigning existing bioinformatic algorithms for a cloud infrastructure to facilitate its use for a wider audience; and (iv) a cloud infrastructure allows a diverse group of researchers to collaborate effectively. The scientific findings will be extended during a follow-up event. Here, we present the applied workflows, initial results, and lessons learned from the hackathon.

## 1. Introduction

While advances in sequencing technology have greatly reduced the cost of whole genome sequencing [1], it has given rise to new problems, especially related to data analysis and management. As the number of bases in the Sequence Read Archive (SRA, [2]) exceeds 33 petabases (June 2019), the difficulty to navigate and analyze all of this data has grown as well. Furthermore, as the number of data warehouses grows with the increased accessibility of the technology, the need to support interoperability of data types increases. To address these issues, the National Institutes of Health (NIHs) launched the Science and Technology Research Infrastructure for Discovery, Experimentation, and Sustainability (STRIDES, [3]) initiative. “Through the STRIDES Initiative partnerships, NIH will test and assess models of cloud infrastructure for NIH-funded data sets and repositories” [3].

As part of the initiative, the National Center for Biotechnology Information (NCBI, [4]) launched a series of hackathons (see biohackathons.github.io), the first of which, the Virus Discovery hackathon, was held in January 2019 in San Diego, CA, USA. These events gather researchers for three days to work on projects around a topic, and provide an opportunity to quickly model a solution or a set of tools to address a community need. NCBI hackathons also facilitate networking among researchers, and allow NCBI staff to identify opportunities to improve their services. While previous hackathons were not specifically focused on working with large volumes of data or compute-intensive tasks in the cloud, they provide a framework with which to engage the research community. As part of the STRIDES initiative, hackathons are particularly focused on allowing researchers to work with large amounts of data in a cloud environment, in an effort to identify the needs and challenges of this new research environment. Typically, topics are developed in conjunction with a host researcher and assigned to different working groups. Team leaders of these working groups are consulted to further refine the topics after some initial recruitment. On the first day of the hackathon, each team discussed its goals and refined their approach before iterating on development goals over the course of the three days. Additionally, one writer from each group participates in a break-out session each day to help guide documentation of the work done.

While there are a number of commercial cloud providers, including Amazon (https://aws.amazon.com), Microsoft (https://azure.microsoft.com), and Google (https://cloud.google.com/), at the time of this hackathon an agreement had been reached with Google as part of STRIDES. Google’s cloud platform offers scalable nodes, with highly configurable access-control settings as well as an SQL-like database infrastructure, BigQuery. Details on cloud infrastructure can be reviewed at [5]. Briefly, cloud compute refers to remotely hosted computers, some parts of which can dynamically access a single compute instance. This access to computing allows research scientists and organizations to use large and scalable computing resources without investing in the required infrastructure and maintenance. While this lowers the barrier to access supercomputer-type resources, it does not provide a comprehensive solution to the general scientific public. The main barriers to adoption by researchers include (i) modest experience in Linux command-line type environments; and (ii) the ineffectiveness of most commonly used bioinformatics tools to leverage the compute resources available in a cloud-computing setting. STRIDES hackathons aim to identify to what extent these barriers impact working researchers, and how they would like them addressed.

One of the fastest growing sources of public biological data is next generation sequencing (NGS) data, housed in NCBI’s Sequence Read Archive [6]. SRA includes results from amplicon and whole genome shotgun studies, conducted on a variety of sequencing platforms. The data is derived from research in many fields such as personalized medicine, disease diagnosis, viral or bacterial evolution, sequencing efforts targeting endangered animals, and sequencing of economically significant crops, among others. Despite the scientific potential of these data sets, there are several impediments to their usage. For one, sample metadata standards can vary between studies, making it difficult to identify data sets based on a common set of descriptive sample attribute terms. Moreover, while the content of some data sets is explicit, this is often not the case, particularly in those samples derived from complex mixtures of organisms like those from the human gut and environmental samples. In these cases, actual organismal content may not be known, either because it was never fully evaluated or because the content includes novel organisms not described in reference sets or otherwise undocumented, i.e., the so-called “dark matter” [7].

Understanding the microbial composition of different environments is necessary to support comparisons between samples and to establish relationships between genetics and biological phenomena. If such information were available for all SRA data sets, it would greatly improve both findability of specific data sets and the quality of analysis that could be conducted. However, determining the organismal content of a sample is not always an easy task as it typically requires comparisons to existing genome references. This can be difficult when samples include viruses because only a small portion of Earth’s viral diversity has been identified and made available in reference sets [8]. Even when content can be identified, the very large size of SRA data sets present a significant scalability problem, and strategies must be developed to support large scale organismal content analysis in order to provide an index of this content for use in data search and retrieval. To that end, the first STRIDES hackathon engaged researchers working in this field to leverage the computational power of the Google cloud environment and test the applicability of several bioinformatic approaches to the identification of both known and novel viruses in existing, public SRA data sets.

Here, we present the general results of these efforts, with an emphasis on challenges participants faced in conducting their work. Firstly, the scientific staff involved in the hackathon is presented, including their demographics and research backgrounds. Scientists were organized in teams, which roughly correspond to the different research sections found in this article, these include: Data selection, taxonomic and cluster identification, domain, and gene annotation.

## 2. Materials and Methods

### 2.1. Participant Recruitment

After initial conception of this project by Ben Busby (BB), J. Rodney Brister (JRB), and Robert Edwards (RE), RE offered to provide a venue for an international hackathon. Participants were recruited through the outreach efforts of BB, RJ, and RE. Vadim Zalunin identified data sets, which were then parsed by RE using PARTIE [9] to look for any potential amplicon or 16 S character, which would be eliminated. The resulting set of SRA runs analyzed in this study can be found at [10].

### 2.2. Assembling Contigs from Metagenomic Data Sets

Contigs containing putative virus sequences were assembled from metagenomic SRA data sets by removing human reads and assembling putative virus sequences into contigs using strategic k-mer extension for scrupulous assemblies (SKESA) [11]. All reads from individual SRA data sets were aligned against the human genome reference sequence (GRCh38.p12) using HISAT2 [12] (see the associated GitHub repository for execution details [13]). Reads mapping fully or partially to the human genome were classified as ‘human’. Putative virus sequences in the remaining read-sets were identified using a *k*-mer taxonomy approach (NCBI, unpublished). Briefly, a subset of 32-mers was sampled from validated sequences in Genbank, and each *k*-mer was assessed for its specificity, with each *k*-mer being assigned to the most recent common ancestor of all the sequences, which were found to contain it. Given a *k*-mer profile for a read-set, the read can be assigned a probability of containing a particular taxon given the frequency of *k*-mers associated with the taxa appearing in the profile. The NCBI viral taxa found to be associated with a read-set using this approach, were used to identify sequences from RefSeq. Given that some viruses are overrepresented in RefSeq, only a few per species were selected at random, while for viruses with segmented genomes, e.g., influenza, all sequences were selected and deduplicated by *k*-mer distances in a later step using MASH [14]. Putative virus reads were assembled using the guided assembler from the SKESA, with the aforementioned RefSeq sequences used as guides, and these contigs obtained identifiers based on the guide accessions with a sequential index as a suffix (for example, the suffix NC_006883.2_1 indicates a contig based on RefSeq Prochlorococcus phage genome, NC_006883.2). In cases where guide selection failed to detect good reference sets, a default viral reference set was used based on the ViralZone database [15].

Reads not classified as virus or human were de novo assembled with SKESA. For the assembled runs, the de novo contigs served as a reference set to align the reads with HISAT2 (as above). The reads that did not align onto either human, viral or de novo contigs were classified as unknown. As a result of the workflow, each run was re-aligned onto human, viral and de novo contigs and contains the same set of reads as the original run. The alignments were converted into SRA format without quality scores and stored in Google cloud storage for later analysis. Given that most SRA metagenomic reads are bacterial or of unknown origin, this step was the most computationally intensive with significant memory and runtime requirements. Due to the limited budget, a timeout was introduced on the de novo assembly step and some runs failed to complete.

### 2.3. Megablast

Contigs and RefSeq virus nucleotide sequences were stored in a single flat file. Coding-complete, genomic viral sequences were extracted from the NCBI Entrez Nucleotide database [16] to create a specific database using the makeblastdb command-line tool [17]. All sequences were compared against all sequences using MEGABLAST [17] with an E-value cut-off of 1 × 10^−10^ and a maximum of one contiguously aligned region (high-scoring segment pair, HSP) per query-subject pair. The average nucleotide identity (ANI) for a pair was calculated as the number of identities divided by the length of the query. Coverage was calculated as the length of the hit divided by the length of the query. For classifications of contigs as known–known, known–unknown, and unknown–unknowns the cut-offs were as follows: Known–knowns, ANI > 0.85 and coverage > 0.8; known–unknowns, ANI > 0.5 and coverage > 0.5; and unknown–unknowns, ANI < 0.5 and coverage < 0.5 or without any hit. Initial heuristics were generated by consulting all of the computational biologists in the hackathon and reaching unanimous agreement.

### 2.4. Markov Clustering

Markov clustering (MCL) [18] was applied to BLAST results as outlined in the associated documentation (https://micans.org/mcl/). Briefly, tabular BLAST output was modified to include only query accession (qacc), subject accession (sacc), and E-value columns, and passed to mcxload to generate network and dictionary files. Thus the set of query and subject pairs is treated as the edge set for a graph, the associated E-values are treated as edge weights. The stream-mirror argument was used to ensure the network was undirected, and stream-neg-log10 and stream-tf ‘ceil (200)’ arguments were used to log transform E-values, setting a maximum value of 200 for edge weights. Finally, the MCL algorithm was run on the loaded network with an inflation value of 10, and 32 threads. All MCL work was performed on a Google Cloud Platform (GCP) machine with 96 cores and 240 Gb RAM.

### 2.5. Domain Mapping

Contigs that were classified by MEGABLAST as ‘unknown–unknowns’ were inspected for domain content using RPSTBLASTN [17]. Briefly, the protein domains from the Conserved Domain Database (CDD) [19] were downloaded and split into 32 different databases, to benefit the most from the available threads. RPSTBLASTN searches were run with an E-value cut-off of 1 × 10^−3^, and the output was generated in JSON format. A working example and the commands used are available on GitHub [20].

### 2.6. VIGA

Modifications were made to the standard VIGA [21] protocol to enhance the overall speed of the program, removing the ribosomal RNA (rRNA) detection step by Infernal [22]. In addition, using the HMMER suite [23], the modified pipeline utilized a combination of the complete pVOGs [24] (9518 HMMs) and RVDB [25] databases to enhance the identification of viral specific hidden-Markov models. Modified scripts and instructions to reproduce all steps are available on GitHub at [26]. All viral annotations were performed on a GCP machine with 96 cores and 360 Gb RAM.

### 2.7. Machine Learning

The Jaccard distance was estimated between the identified viral contigs using MASH [14], a technique shown to be a reliable tool to cluster amplicon data sets, metagenome read data sets, and contig data sets [27]. A *k*-mer size of *k* = 21 bp was chosen with a sketch size of 10,000. Samples containing less than two viral contigs were removed from the analysis. A total of 511 samples were kept for the analyzed and clustered by Ward clustering. A manual cleaning of the terms was performed to remove punctuation and low-informative terms. In total, 210 samples with abstract and comments were analyzed. Partial least squares regression was performed using pls (https://github.com/bhmevik/pls), with eight components and using the regression algorithm. SRA metadata was analyzed using Doc2Vec, as implemented in the Python package gensim (https://radimrehurek.com/gensim/models/doc2vec.html). The model of the metadata was trained over 10 iterations, with an initial learning rate of 0.025, and a minimum rate of 0.0025. The feature vector size was set to 300. This high dimensional model was reduced via application of t-distributed stochastic neighbor embedding (tSNE) as implemented by the Python package scikit-learn [28]. The model was reduced to two dimensions, over 300 iterations, with a perplexity of 40.

## 3. Results

### 3.1. Hackathon Planning and Preparation

The number of metagenomic data sets in the SRA database is steadily increasing [29], albeit not all the information that each SRA contains has been exploited to the fullest, e.g., not all species within sequencing data sets are routinely identified. A major hurdle for a detailed analysis of metagenomic data sets is the lack of readily available hardware and analysis pipelines. The goal of this hackathon was to identify user needs for standard NGS data analysis in a cloud environment as it can offer more computational power than is available on local processors. Viruses are present in virtually all organisms and environments, but only a limited number have been identified so far [30]. Therefore, virus sequences present a suitable model for NGS data mining.

Robert Edwards at San Diego State University agreed to sponsor the event and 36 participants registered for the event. Participant demographics are outlined in Table 1. Participants came from a variety of academic backgrounds, worked at a variety of institution types and at all stages of their career from training to senior investigator. The wide range of backgrounds allowed us to get a broad perspective on the hurdles faced by researchers working in a cloud environment.

After participants had been identified, teams were developed and team leaders identified. The team leaders were invited to an online, pre-hackathon event to orient them to the data and working in a cloud environment. This also allowed us to further refine the scope of the event and the focus for the various groups. At this event a data selection strategy was settled upon and a general approach was decided upon for most of the other groups (outlined in Figure 1). Unlike in a typical NCBI hackathon, the groups were not working on independent projects, but were instead dependent on the work of “upstream” groups. At the time of the hackathon, an agreement for data hosting had been reached only with Google, and so data was uploaded to the Google cloud environment; all data uploaded was already publicly available via the NCBI’s SRA resource. The data chosen to be uploaded for this event, along with the selection criteria, is described in the following section.

The pre-hackathon event highlighted the need for more of an introduction to doing bioinformatics in the Google cloud environment, as well as an opportunity to improve workflows by pre-installing common tools on the virtual machines (VMs) to be used by “hackathoners”, both of which were addressed before the actual event. The documentation developed can be found at the GitHub repository linked above, though it will likely need to go through more revisions to adequately address the needs of researchers new to working in a cloud environment. The pre-installed tools are outlined in Appendix A.

Jupyter was found to be a popular environment to work from, and was not preinstalled. Work to identify the best Jupyter-style solution to a hackathon’s needs is ongoing, and includes exploration of GitHub’s Binder, Google’s CoLab, and custom Jupyter and JupyterHub [31] set-ups. Having a dedicated IT support person on site was immensely helpful, for the various technical issues that are bound to arise at this type of event, but also to facilitate launching VMs as necessary for participants and adjusting hardware specs as necessary. Of note, it was found to be important to launch instances with a fixed external internet protocol address (IP) to prevent confusion when an instance is taken down and relaunched for any number of reasons.

### 3.2. Data Selection

The goal of the data selection step was to determine how many of the Short Read Archive (SRA) data sets are linked to Whole Genome Shotgun (WGS) projects in order to include all publicly available metagenomic studies, and exclude amplicon sequencing studies that would not include viral sequences. The records of all 141,676 data sets in SRA were processed using PARTIE [9] and identified 85,200 SRA entries as WGS studies. During the hackathon, we subsampled the total SRA entries into a smaller test data set made up of 2953 samples. These ~3000 samples would be used for all further analyses. The ~3000 samples were selected from the complete data set in three ways: Randomly (*n* = 1000 samples), based on the size of the data set (*n* = 999 samples), and large data set with a relatively high percentage of phage content (*n* = 999 samples). A complete list of SRA accession numbers for each category can be found in our GitHub repository (https://github.com/NCBI-Hackathons/VirusDiscoveryProject/tree/master/DataSelection).

From this smaller data set, approximately 55 million contiguous assemblies (contigs, hereafter) were assembled. As many of these samples represent complex metagenomes containing multiple organisms and viruses, only about half of the raw reads were assembled into contigs. The participants were pleased with the contigs, and found them a useful way to get insight into the genomic content of individual SRA data sets. That said, there was interest in exploring the suitability of different assemblers for this task. Given the heterogeneity of SRA data, preselecting the data was critical to the success of the event. However, given that the groups were not working on independent projects and that they were not all familiar with working with such a volume of data, the event might have been improved by identifying a much smaller subset for development and testing. Further, pre-selecting data sets suitable for each group would have alleviated some of the issues associated with the groups being dependent on each other’s work.

### 3.3. Data Segmentation

To improve the processing speed and to identify contigs harboring a putative virus sequence we filtered the 55 million contigs into three distinct categories. The 55 million contigs from the data selection step were pre-filtered based on size by removing all contigs shorter than 1 kb in length. The remaining 4223,563 contigs were then screened by BLASTN [17] against the virus RefSeq database [32] using a cut-off e-value of ≤0.001 and classified into three categories based on the average nucleotide identity (ANI) and alignment coverage (Figure 2):

Known–knowns: 12,650 contigs with high similarity to a known RefSeq virus genome, with ANI >85% and contig coverage >80%. These contigs showed similarity to bacteriophages. In particular, 19 bacteriophage species showed hits to more than 100 contigs and, specifically, crAssphage [33] comprising ~27% of all known–known contigs.

Known–unknown: 6549 contigs moderately similar to known viruses. This category was further divided into two subcategories. The first category contains 4713 contigs with high similarity to known viruses (>85% ANI) but the alignment covers between 50%–80% of the contig length. The second category contains 1836 contigs with a lower alignment similarity to known viruses (50%–85% ANI) and cover >50% of the contig length. These likely belong to either somewhat distant relatives of known viruses, close relatives that have undergone recombination other viruses, known viruses with structural variants, or prophages where the regions that did not align to the RefSeq viral database correspond to the bacterial host.

Unknown–unknown: 4,204,364 contigs with BLASTN hits that did not meet the criteria for ‘known–knowns’ or ‘known–unknown’, as well as any contigs where no hits were found. These contigs comprised the vast majority of processed contigs.

Cellular sequences from RefSeq were not utilized to filter out non-viral sequences as most cellular genomes contain one or more proviruses/prophage [34]. The ‘unknown–unknown’ contigs from metagenomic samples that did not undergo virus enrichment are likely primarily bacterial and other cellular sequences. However, many novel viral sequences can be expected in this set. Some of the BLAST results were pre-computed and loaded into Google’s BigQuery, to provide a reference for initial testing during the hackathon. Many of the hackathon participants were not familiar with a large scale analysis from databases, and better tutorials on how to leverage cloud infrastructure may be warranted. Therefore, while SQL-like tables may be a convenient means of presenting data, some additional training is necessary for them to be useful. Finally, while salient viral information was obtained from thousands of contigs, the observation that a vast majority of the contigs could not be characterized by this method underscored the need for fast domain-mapping approaches to categorize this type of data.

### 3.4. Data Clustering

To reduce the size of ‘unknown–unknown’ data for subsequent analysis steps, we aimed to identify sequence clusters, which could be subsequently analyzed faster due to their smaller size. To this end, all virus RefSeq sequences and the contig sequences were aligned against each other by combining them into one BLAST database. This self comparison of 4,223,563 contigs yielded 245,652,744 query-subject pairs, which were treated as edges of a graph with edge weight equal to the log of their E-value. The graph was then clustered via Markov Clustering (MCL) [18], and the resulting subgraphs, or clusters, analyzed. The distribution of cluster sizes is seen in Figure 3. A total of 2,334,378 clusters were returned, representing an approximately two-fold reduction in data. Of these clusters, 57% (1,331,402) were singletons (i.e., a cluster size of 1), indicating that the contig was unique among those contigs analyzed.

One challenge with this approach is the computational resources required and the poor scaling with sample size. Additionally, interpreting such a large volume of BLAST results is non-trivial, and even MCL took 12 h to cluster the results with 32 cores and 240 Gb RAM (interestingly, MCL was found to be unable to effectively take advantage of the full 96 cores available); thus, when working in the cloud with big data, additional tools to support the analysis of the results of traditional tools (when applied at a large scale) will be beneficial.

### 3.5. Domain Mapping

To gain more information about the ‘unknown–unknowns’ contigs, which can harbor putative novel viruses, we screened them for conserved domains as domains can identify contigs with virus related features, e.g., capsids, but were too diverse for detection using sequence alignments. In order to get a more nuanced assessment of the genomic content of these contigs, we attempted to align 4.2 million contigs against the Conserved Domain Database (CDD) [19]. The entire CDD database was split into 32 parts in order to benefit most from the available threads, and the contigs were analyzed via RPSTBLASTN [17] against the sub-sectioned database. Domains with significant hits (E-value < 0.001) were subsequently divided into five groups containing the corresponding position-specific scoring matrices (PSSMs), based on their CDD accession number. These groups were created by using the ‘organism’ taxonomic information provided with CDD, resulting in a viral group (2082 PSSMs), a bacterial group (19,383 PSSMs), an archaeal group (1644 PSSMs), a eukaryotic group (17,201 PSSMs), and a so-called unknown group (15,682 PSSMs). To reduce the computational burden downstream, contigs have been filtered based on the taxon-specific PSSMs they carried. Contigs that carried no viral hits and more than three hits to prokaryotic or eukaryotic CDDs or carried more than three eukaryotic CDDs were excluded from the further analysis.

We were able to process 347,188 contigs out of the 4.2 million contigs annotated as ‘unknown–unknowns’. Out of those 347,188 contigs, 180,820 (52%) were excluded based on the criteria above, and 166,368 passed for downstream analysis (48%). Out of those contigs that were passed to the domains group, 39,986 (12%) were classified as ‘dark matter’, i.e., having no hit to any CDD. Most of the excluded contigs had more than three bacterial CDD and no viral CDD hits. Overall, subjected contigs had enrichment for both bacterial and unknown PSSMs in comparison with the other three categories (Figure 4). This could be due to the overrepresentation of these PSSMs in the database, but since there was a comparable number of eukaryotic CDDs present, this skewness was more likely a reflection of the input data.

In the set-up of the analysis, we chose the JSON [35] format, since this would be the easiest way to incorporate downstream in the index (vide infra). However, the algorithm itself does not allow specification of what exactly is included in this format. Therefore, the output is unnecessarily bulky and quickly becomes more than cumbersome to work with. Output flexibility would vastly increase the potential of this output format for this amount of data.

### 3.6. Gene Annotation

To better classify contigs as putative virus sequences, we performed an additional step to identify open reading frames (ORFs) and map virus specific proteins to contigs with already identified domains. Putative viral contigs were characterized using a modified viral annotation pipeline, VIGA [21]). Briefly, the contigs have their ORF predicted with Prodigal [36] and annotated against RefSeq viral proteins with BLAST [17] and DIAMOND [37]; and search for conserved motifs from pVOGs [24] (prokaryotic virus) and RVDB [25] (all virus-like sequences but not from prokaryotic viruses) using HMMER [23].

Tackling a very large data set, computational efficiency was a concern. While BLAST and DIAMOND can be parallelized to a certain degree, i.e., using all cores/threads from a single computing node but not across several computing nodes, HMMER lacks an efficient multithreading parallelization (http://eddylab.org/software/hmmer/Userguide.pdf). To partially mitigate this behavior, VIGA was parallelly invoked from the command line to run as many instances as processors (CPUs) asked, instead of a single instance with all CPUs (https://github.com/NCBI-Hackathons/VirusDiscoveryProject/blob/master/VirusGenes/scripts/director.sh to consult the director script). Each VIGA process was started with only the contigs from a single SRA data set on a single processor, and later ran 160 such processes in parallel. Initial test runs of 4400 contigs running on 160 processors showed performance of about 25 sec/contig/processor. In real-time, one million contigs will take approximately 7000 processor hours. Results from the modified VIGA pipeline provide viral-specific taxonomic/functional annotations to all putative viral contigs (Figure 5), based on a similarity search by sequence alignment (BLAST and DIAMOND) and modelization (HMMER against pVOGs and RVDB). Virus contig IDs are appended to the VIGA output and putative protein sequences were extracted from the GenBank output. All information from the modified VIGA pipeline is codified into a hierarchical JSON format for downstream processing and storage (https://github.com/NCBI-Hackathons/VirusDiscoveryProject/blob/master/VirusGenes/scripts/converter.py).

As noted above, processing such a large volume of data requires massive parallelization, a task that occupied a significant portion of this group’s time. Relatedly, interpreting the volume of results provided remains a challenge. Different algorithms may have different computational costs or needs (CPU- vs. memory-expensive process). Therefore, successful pipelines should fine-tune those needs to the available resources. Combination of different search strategies increases the run time of the pipeline but, if run under an appropriate decision tree, increases the confidence during taxonomic and/or functional annotation.

### 3.7. Metadata Analysis

As missing metadata often complicates identifying NGS data sets of interest and interpreting the results of analyses of public SRA data, we tried to infer metadata information based on SRA data set contig content. SRA runs were clustered using MASH [14], and six main clusters of samples were identified, showing certain diversity in terms of viral content across the data set. In order to unravel the drivers of that composition-based clustering, the words from the SRA study comments and abstracts were extracted using SRAdb [38]. A vector of word frequencies was constructed across the selected samples. A partial least squares regression (PLS) was performed in order to identify any co-variance between the identified clusters using MASH and the word frequencies associated with the samples. No strong co-variance could be identified using this approach, suggesting that abstracts and comments vocabularies are too vague to automatically characterize samples, Figure 6A.

As a proof of concept, we show that natural language processing (NLP) trained on SRA data and associated project metadata can identify individual SRA data sets from human gut microbiome metagenomes. Doc2vec [39], an NLP algorithm that uses unsupervised learning to embed variable-length texts into a vector, was trained on the SRA metadata of 628 samples and transformed the metadata into a 300-dimension vector. t-SNE [40], a popular dimensionality reduction tool, was trained and transformed the vectors into coordinates for a 2D space. The SRA metadata was labeled based on the “center_project_name”, which is typically used to identify the environment from which the metagenome was sequenced from. Three “center_project_name” classes were examined: “human gut microbiome”, “NA”/“None”, and “other”. Figure 6B shows that all three classes are easily and cleanly separable. Some possible uses of this technique could be to train a classifier with this data set and correct mislabeled metadata or annotating SRA data sets with missing metadata.

## 4. Discussion

Here we presented the results from the NCBI’s Virus Discovery Hackathon. A diverse group of international researchers met and characterized the viral content in ~3000 metagenomic SRA data sets, and in doing so, identified opportunities to apply bioinformatic approaches using cloud computing infrastructure to the analysis of NGS data sets. The original intent of the hackathon was to develop an index of SRA run sets that could be searched based on the viral content contained within the runs. To that end, several use cases were identified to guide development.

The use cases developed are outlined below. (i) Identifying shared genomic content across runs. Thus users may submit a sequence, and find all runs from which similar contigs can be derived. (ii) Filter based on run metadata. This is essentially the same service provided by the NCBI Entrez Index. (iii) Gene/domain based searches. Users may want to find only runs likely to encode some gene or functional domain of interest, as determined by an analysis of contigs assembled from the runs. (iv) Searching based on virus taxonomy. A user may want to find runs likely to contain a particular viral taxon based on an analysis of contigs. These goals informed the analyses performed.

Despite generating a number of interesting insights, technical challenges prevented more rapid progress. That said, we feel that these represent opportunities for future development to enhance cloud-based bioinformatic infrastructure and practice. While everyone involved appreciated working with contigs, as opposed to the reads, the sheer volume of SRA data means that the contigs do not represent enough data compression for efficient workflows. While effort was made to identify a test data set, this data set was still perhaps too large, as it represented nearly 55 million contigs. Thus, for future hackathon-type events, especially if the focus is on big data, it is recommended that a number of test sets be developed of various sizes, ideally nested such that the smaller sets are subsets of the larger sets, and that they capture the diversity of the full data set as much as is possible. Clustering proved to be a promising data compression approach, but further work needs to be done to both improve the efficiency of the algorithms when working at this scale, and to determine the content of the clusters generated. Of particular note, the large number of singletons identified is worth investigating to determine if they represent artifacts of either the sequencing or assembly strategies or if metagenomic data really contains such a large number of unique sequences.

Of the classifications made, it is worth noting that the most abundant extant virus in the data sets analyzed was crAssphage (Figure 2C). While this may reflect the pre-selection strategy employed (WGS metagenomic studies were targeted), it underscores the prevalence of this recently identified virus. Further, as seen in Figure 2B, majority of viruses with hits to the RefSeq virus database could be accurately assigned to previously discovered viruses (known–known classifications). This suggests that the RefSeq set is a good representation of extant viral sequence space. However, as seen in Figure 2A, the majority of contigs lacked any hits to the RefSeq virus data set. This may be due in part to the exclusion of non-viral RefSeq from the comparison. Indeed, as seen in Figure 4B, there was likely an abundance of bacterial sequence contained in these data sets. Still, the even larger set of sequences with domains of unknown function supports the claim that dark matter forms a large portion of all sequencing data. Thus, improved methods for classifying sequences without any close hits among extant sequence space are needed.

Jupyter [31] was immensely popular as a framework from which to develop workflows and conduct exploratory analysis. However, supporting Jupyter in the cloud is not straightforward. Simultaneously supporting collaboration between groups, controlling access to machines, and allowing access to data buckets is challenging. Further efforts are needed to determine which notebook formats are best suited to the hackathon environment. Relatedly, it was found that, when working at such a large scale, input/output (I/O) remains a hurdle and workflows developed around big data analysis in the cloud should accommodate this. Another challenge, felt most acutely by those working on applying machine learning to SRA data, is the need for clean metadata. While it was found that analysis of SRA metadata could separate samples according to their center_project_name attribute, it is worth noting that this field in the biosample does not use a controlled vocabulary, and so it is unclear what the difference between the “NA”/” None” and “other” actually represents. When we spend time curating data sets we should work on the ones with the most metadata, and this should be considered when constructing test data sets in the future. Additionally, it was found that not all data labeled as WGS appeared to be WGS data, emphasizing the need for better metadata documentation by the research community. The sharing and reuse of data is one of the primary drivers behind open, FAIR (Findable, Accessible, Interoperable, Re-usable) bioinformatic cyberinfrastructure [41].

As discussed above, many SRA entries have incomplete metadata, which deters researchers from performing their own analyses on other scientists’ data. Completing the metadata would promote the reusability of data archived in NCBI’s databases.

A major goal of this work was to establish domain profiles of NGS data sets, as these have immense potential for supporting sorting and filtering of these massive data sets. They should be treated as first-class reference objects, and a massive expansion of these data objects may be the most effective way to expand into new data spaces. To this end, a follow-up hackathon is currently being planned, during which it is hoped that progress can be made on identifying a Jupyter framework that supports collaborative pipeline development, and which will result in an index of at least a small portion of the metagenomic data set available in the SRA.

## 5. Conclusions

Conservatively assembled contigs support initial exploration of SRA data.Redesigning algorithms to leverage cloud infrastructure would make cloud environments more accessible to a wider audience.Approaches to classifying—and reporting the classification of—contigs can be effectively developed via collaboration between a diverse group of researchers.Analysis will continue in a follow-up hackathon.

## Figures and Tables

**Figure 1 genes-10-00714-f001:**
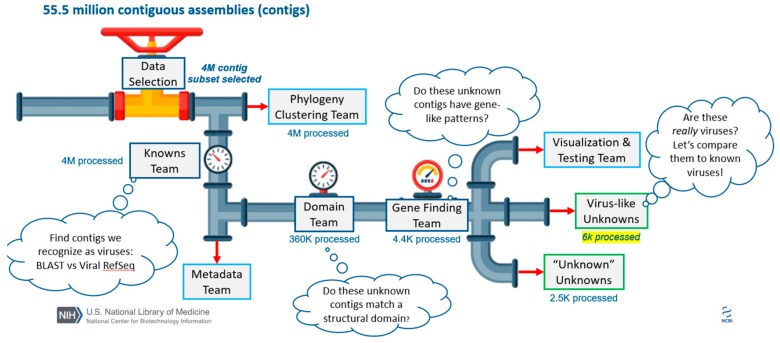
Overview of hackathon teams and data processing. All numbers detail the number of contigs processed at each step of the pipeline. A subset of ~3000 data sets were assembled, generating 55.5 million total contigs. Researchers attending the hackathon assembled into teams that roughly correspond to goals outlined in the Methods and Results. Members of the “Knowns Team” excluded contigs based on size (removing <1 kb in length) and the remaining ~4 million contigs were assigned classification to known viruses using a BLASTN search against the RefSeq Virus database (Section 2.3 and Section 3.3). Independently, members of the “Phylogeny Clustering Team” clustered ~4 million contigs using Markov Clustering techniques (Section 2.4). Members of the “Metadata Team” used machine learning approaches to build training sets that could be used to correlate sequences to sample source metadata (Section 2.7 and Section 3.7). Members of the “Domain Team” predicted functional domains with RPSTBLASTN and the CDD database using ~360,000 contigs that were not classified using the RefSeq Virus database (Section 2.5 and Section 3.3). Members of the “Gene Finding Team” predicted open reading frames and putative viral-related genes using the modified VIGA pipeline on ~4400 putative viral contigs (Section 2.6 and Section 3.6). Members of the “Visualization Team” devised ways to display complex data and the “Testing Team” accessed if components of the pipeline were accessible to future users. Two additional teams were tasked with analyzing sequences, which could not be identified as confidently cellular or virus-like with the methods described above (Section 3.5).

**Figure 2 genes-10-00714-f002:**
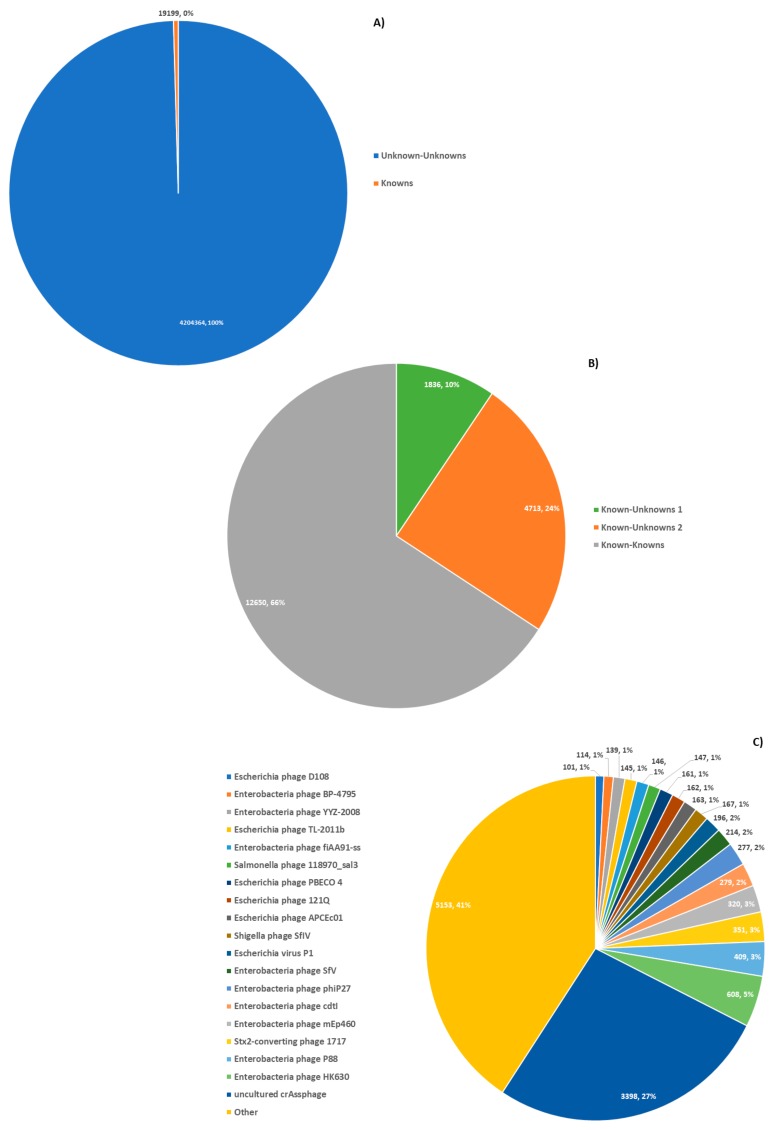
Taxonomic classification of contiguous assemblies. Contiguous assemblies were compared to Virus RefSeq via BLASTN; (**A**) >99% of contigs have no hits amongst RefSeq viruses. (**B**) Of those with hits against RefSeq viruses, the majority had close matches, known–knowns. The remainder either had a weak match to a RefSeq, ‘known–unknowns’ 1, or a strong match to a RefSeq subsequence, ‘known–unknowns’ 2. (**C**) Top RefSeq viruses matched by ‘known–known’ contigs, by the number of contigs with a match to them. CrAssphage represents the most abundant virus in the subset.

**Figure 3 genes-10-00714-f003:**
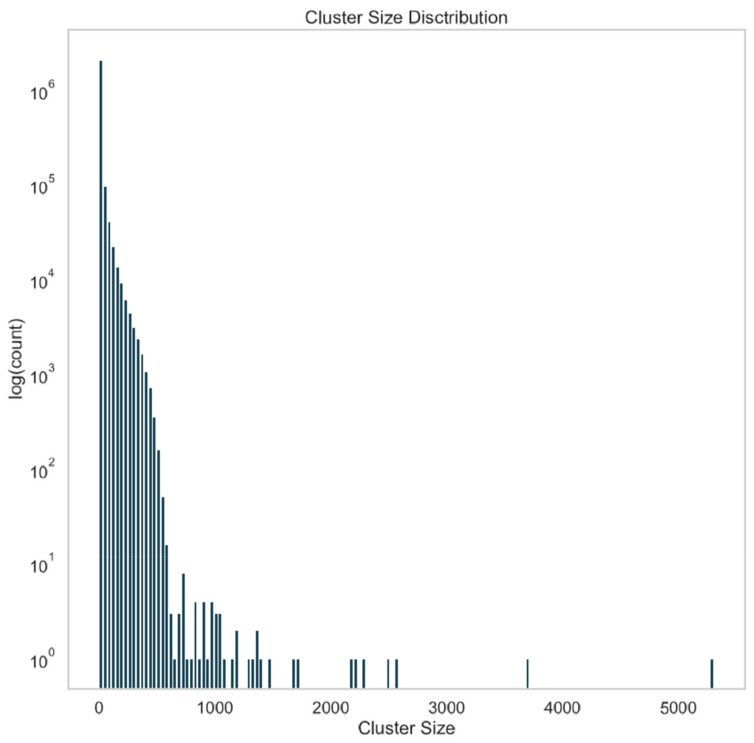
Abundance of contig clusters by cluster size. All of the contigs with length greater than 1 kb were combined with the RefSeq viral data set. An all-vs.-all comparison was made via BLASTN. BLAST hits were treated as edges in a graph with a weight equal to the log transform of the E-value, and this graph was clustered using Markov clustering. The resulting clusters were then analyzed for size, and a histogram of cluster size is shown. Approximately half of the clusters are singletons.

**Figure 4 genes-10-00714-f004:**
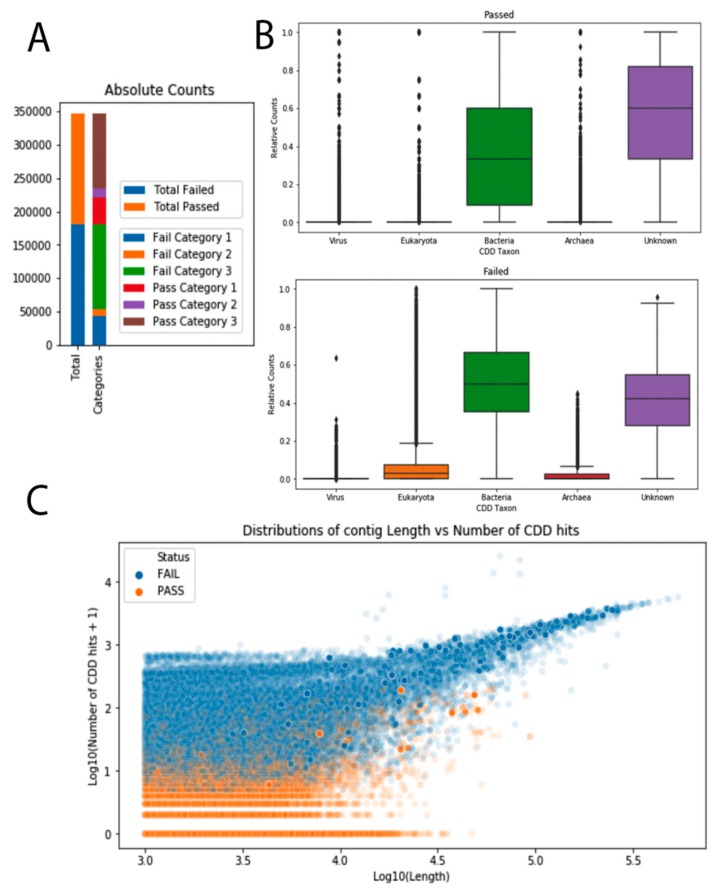
Overview of the results obtained by classifying unknown–unknown sequences with RPSTBLASTN; and (**A**) absolute counts for the total contigs analyzed. On the left the total amount of passed and failed contigs are shown. On the right, “passed” and “failed” groups are split into three categories. Fail category 1 is the group of contigs that carried more than three eukaryotic Conserved Domain Databases (CDDs), more than three bacterial CDDs, and zero viral CDDs. Fail category 2 is comprised of contigs with more than three eukaryotic CDDs. Fail category 3 contains contigs that have more than three bacterial CDDs and zero viral CDDs. Pass category 1 contains the so called dark matter group (no CDD hits at all). Pass category 2 contains contigs that have more than zero viral domains, and pass category 3 contains all the rest of the passed contigs. (**B**) Relative counts for the five different taxonomic bins of the CDDs are shown for the passed group of contigs (top) and the failed group of contigs (bottom). To account for contig length, the number of CDD hits per category was divided by the total number of CDD hits in one contig. (**C**) Distributions of contig lengths vs. number of CDD hits for failed (blue) and passed (orange) contig groups. Both the contig length and the number of CDD hits are log-transformed.

**Figure 5 genes-10-00714-f005:**
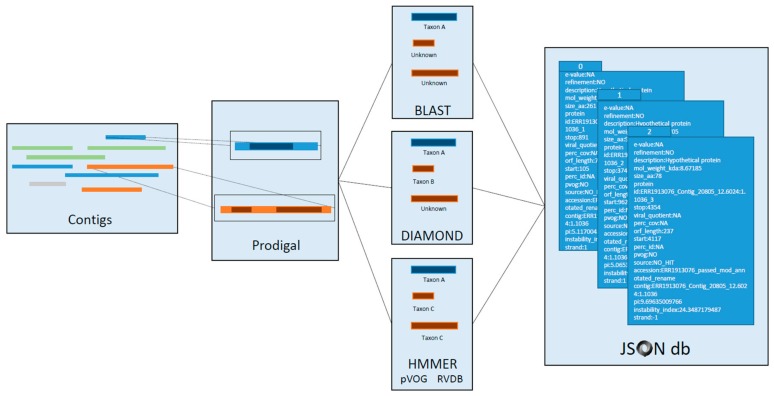
Sequence annotation with VIGA. Putative viral sequences were annotated using the modified VIGA pipeline. Contigs depicting (putative) viral sequences, prefiltered by the presence of a viral CDD, were passed to VIGA. Each contig had their open reading frames (ORF) detected and translated with Prodigal using the 11th genetic code. Each viral protein was further characterized with a combination of three different annotation methods: BLAST, DIAMOND, and HMMER. HMMER included two model databases: pVOGs (prokaryotic viruses) and RVDB (all viral sequences but prokaryotic viruses). Coordinates, protein translation, hits, E-values, viral quotient, percentage of identity, and other meaningful information are codified in a hierarchical JSON database for downstream analysis.

**Figure 6 genes-10-00714-f006:**
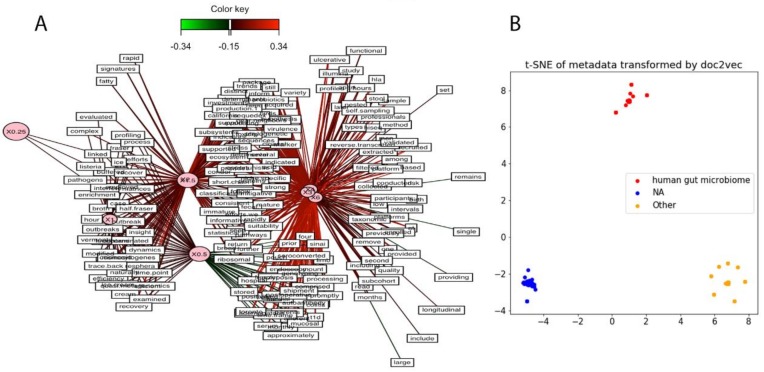
Next generation sequencing (NGS) classification using associated metadata. (**A**) Study abstract and metadata are insufficient for NGS classification. Data sets where clustered using MASH, and partial least squares regression was performed to identify any covariance between the sequence content and word frequencies derived from the associated study abstracts and metadata; and (**B**) human gut microbiome samples are separable from other studies using Sequence Read Archive (SRA) metadata. Metadata was used as input to a word2vec model with 300 features, and the model was reduced to two dimensions using t-distributed stochastic neighbor embedding.

**Table 1 genes-10-00714-t001:** Participant Demographics. Demographic summary of hackathon participants.

	Raw	% Participants	% Responses
Participants	37	NA	NA
Survey Responses	36	97.3	NA
Institutional Affiliation			
Academic	27	72.97	75.00
Government	6	16.22	16.67
Other	2	5.41	5.56
Unknown	1	2.70	2.80
Educational Attainment			
Ph.D.	14	37.84	38.89
M.S.	12	32.43	33.33
B.S.	2	5.41	5.56
Unknown	8	21.62	22.22
Career Stage			
In Training	11	29.73	30.56
Junior	14	37.84	38.89
Senior	6	16.22	16.67
Unknown	5	13.51	13.89
Programming Language			
Shell	33	89.19	91.67
Python	31	83.78	86.11
R	26	70.27	72.22
Perl	13	35.14	36.11
Java	10	27.03	27.78
C/C++	9	24.32	25.00
JavaScript	4	10.81	11.11
SQL	3	8.11	8.33
Matlab	2	5.41	5.56
Other	4	10.81	11.11

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
