# Peer review of "NCBI’s Virus Discovery Hackathon: Engaging Research Communities to Identify Cloud Infrastructure Requirements"

_genes, 2019, doi:10.3390/genes10090714_

Round 1

Reviewer 1 Report

This is an interesting manuscript which discusses attempts made to tackle a known problem, the difficulty in large scale analysis of publicly available high throughput sequencing datasets and, more specifically, the challenge of identification of viral transcripts in this data.

Many of the conclusions are useful and would be relevant both for researchers in the field of virus discovery and for those planning future genomics hackathons.  A large number of widely used bioinformatics tools and their usefulness in this context are assessed. The shortfalls in the current frameworks available for cloud-based bioinformatics are also discussed. I have a few minor points for consideration as follows.

The direction of the manuscript is quite muddled in a few places - it is not clear whether the intention is to provide advice for researchers organising further big data hackathons, to present a set of curated datasets (as specified in the abstract), to review the use of cloud infrastructure in bioinformatics, to present a pipeline for virus discovery or to review different approaches to the steps in a virus discovery pipeline.  It would improve the manuscript to focus on one or two of these objectives or to take them one at a time, rather than to focus on all simultaneously. 

I’m not sure if the problem is with the original submission or the processing of the submission but the image quality is very poor and in several cases the text is not visible.

The materials and methods section is not ordered very well - it is not completely clear which steps were performed in which order.   A line at the beginning of each section with the objective of the step would also be useful (e.g. in order to differentiate between human and viral contigs….., in order to identify known viruses…..) as the technical details don’t always make it clear what the analysis was designed to do.  Some parts of the materials and methods also lack sufficient detail to reproduce the analysis without extensive reference to the associated github repository, the presentation of which is quite confusing. A large number of datasets are mentioned in the abstract but later in the text several subsets of this data are discussed and it should be made clearer which analyses were performed on which subset and why this subset was selected.

I also have specific comments on the text as follows:

Line 46: What is a pipeline of datasets?

Line 50: A large proportion of these 141,000 were not used during the hackathon so it may be misleading to mention them here

Line 79: I’m not sure what “the teams further scope directions” means

Line 88: Reviewed where?

Line 145: More detail is needed about this approach if it is unpublished - identifying the putative viral reads is fundamental to this study

Line 146: Remaining after what?

Line 149: Which guide accessions were used?

Line 164: This is the first mention of BLASTN in the M&M - what were the criteria to be classed as unknown-unknowns?

Lines 197 - 212: It is not completely clear from this paragraph what these steps were for

Line 225: I don’t think I have a Table 1 - I have Supplementary Table 1 but this is a list of software.

Line 268 - 277: This paragraph is quite confusing - I’m not very clear on how the original SRRs were selected, what the “whole genome sequences” database is or which steps were performed only on the smaller test dataset.  

Line 272 - 274: What do the numbers in brackets represent?

Line 275: What are the categories?

Line 279: Why does it only represent half the raw reads?

Line 294: Did all of them show similarity to bacteriophage or some were similar to other viruses?

Lines 310-312 and 319-321: This sentence is there twice

Lines 331 - 348: Was one contig then selected to represent each cluster?

Line 346: I’m not sure if this is an issue with BLAST, but rather with clustering a large number of data points - a clustering step would be needed regardless of the algorithm used for the pairwise comparisons.

Line 400: Which version of BLAST was used which allows parallelisation?

Line 412: “virus hunting tool kit” needs a reference

Line 453: What is the difference between “NA” and “other”?  If NA means unannotated it is surprising that these are so clearly distinct from the “other” category.

Line 454: How could this technique be used to correct mislabeled metadata or replace missing metadata when it requires the metadata to classify the sample?

Line 495: This is not very clear - does it mean most of the hits against RefSeq were good quality hits?

Figure 1

The use of team names in this figure make it quite confusing, for example the “phylogeny team” and “visualisation team” are not mentioned elsewhere in the text. It is not clear what each step in the pipeline aims to achieve. It would be helpful to specify the role of each team as well as the name.

Figure 2

The labels on the figure are too small to see in this version of the document.  Several of the colours in pie chart C are reused so it is not clear which is which - a bar chart might show this data more clearly.

Figure 4

I’m not sure what is shown in 4B - counts relative to what?  If each contig was classified into one category what are the distributions here?

Figure 6

The text is too small to see in this version of the document.  

Author Response

Overall 

We rearranged and clarified the manuscript in several places to address the issues raised by both reviewers concerning the clarity and purpose of the manuscript. This includes extending the Abstract, improving and clarifying Figure legends and adjusting references. The quality of the figures was improved. We were surprised about the missing Table 1, which was included in our initial submission.  

Reviewer 1

The direction of the manuscript is quite muddled in a few places - it is not clear whether the intention is to provide advice for researchers organising further big data hackathons, to present a set of curated datasets (as specified in the abstract), to review the use of cloud infrastructure in bioinformatics, to present a pipeline for virus discovery or to review different approaches to the steps in a virus discovery pipeline. It would improve the manuscript to focus on one or two of these objectives or to take them one at a time, rather than to focus on all simultaneously.

Response: We extended the abstract, added short summary sentences  to each result section, and rephrased several paragraphs in the manuscript  to clarify the main points of the paper (see below and related Reviewer 2 comment). 

I’m not sure if the problem is with the original submission or the processing of the submission but the image quality is very poor and in several cases the text is not visible.

Response: We are also concerned that high quality versions of our figures were not received by the reviewers. Figures 2 and 6 were also reworked to improve legibility of the labels.

The materials and methods section is not ordered very well - it is not completely clear which steps were performed in which order.   

Response: We reordered the Material and Methods section to reflect the order of Results  section. 

A line at the beginning of each section with the objective of the step would also be useful (e.g. in order to differentiate between human and viral contigs….., in order to identify known viruses…..) as the technical details don’t always make it clear what the analysis was designed to do.  

Response: We thank the reviewer suggesting this. We added a short summary at the start of each Result section explaining the reason and purpose.

Some parts of the materials and methods also lack sufficient detail to reproduce the analysis without extensive reference to the associated GitHub repository, the presentation of which is quite confusing.

Response: Many parts of the materials and methods section were reworked to address this and other concerns (below). We updated the GitHib Readme with information which sections in the publication related to which repository folders.

 A large number of datasets are mentioned in the abstract but later in the text several subsets of this data are discussed and it should be made clearer which analyses were performed on which subset and why this subset was selected.

Response: We clarified the datasets and their origin in the abstract and in each result section. In some cases, the changes were the results from addressing  other comments.

Minor:

Line 46: What is a pipeline of datasets?

Response: We agree with the reviewer that this as an unclear term and adjusted it accordingly. The part in question reads now “crowdsourced set of analysis and processing pipelines for complex biological dataset“

Line 50: A large proportion of these 141,000 were not used during the hackathon so it may be misleading to mention them here

Response: We agree with the reviewer that the initial sentence could mislead readers. However, the 141,000 were used to create the data set analyzed during the Hackathon. Therefore, we mention them in the abstract but stress they were not part of the Hackthon. 

Line 79: I’m not sure what “the teams further scope directions” means

Response: We agree with the reviewer that the term was unclear. We clarified the sentence as follows: “On the first day of the hackathon, each team discussed its goals and refined their approach before iterating on development goals over the course of the three days”

Line 88: Reviewed where?

Response: We thank the reviewer to point us to this link. In the initial  manuscript this was a named link which got lost during the submission. The corresponding book has been added as a reference ([5]).

Line 145: More detail is needed about this approach if it is unpublished - identifying the putative viral reads is fundamental to this study

Response: We agree with the reviewer that more details are required. The approach was elaborated upon and extended.

Line 146: Remaining after what?

Response: We agree with the reviewer that this sentence was unclear. It related to the taxa "remaining" after k-mer based taxonomy assessment of the run-set. This section was reworked to improve clarity.

Line 149: Which guide accessions were used?

Response:  We used RefSeq sequences from the species identified via the k-mer taxonomy approach as being prevalent in the data set. The section was reworked to make this point more clear.

Line 164: This is the first mention of BLASTN in the M&M - what were the criteria to be classed as unknown-unknowns?

Response: We agree with the reviewer that the criteria were unclear as it was.  The blast methods were moved prior to this section and the classification strategy was outlined.

Lines 197 - 212: It is not completely clear from this paragraph what these steps were for

Response: This section was reworked to improve clarity.

Line 225: I don’t think I have a Table 1 - I have Supplementary Table 1 but this is a list of software.

Response: We are unclear why the reviewers did not receive this table, as it is included in our versions of the manuscript.

Line 268 - 277: This paragraph is quite confusing - I’m not very clear on how the original SRRs were selected, what the “whole genome sequences” database is or which steps were performed only on the smaller test dataset.  

Response: The confusion is on our end. This paragraph has been cleaned up to correctly reflect the source of the datasets and which samples were further processed. WGS should be referring to the Whole Genome Shotgun database at NCBI. We have highlighted which samples were processed during the event. And avoid confusion by mentioning processing of the starting dataset.  

Line 272 - 274: What do the numbers in brackets represent?

Response: The number in parentheses represent the number of samples in the subsampled dataset meeting the described criteria. We added “samples” to each number for clarity.

Line 275: What are the categories?

Response: We clarified them in the paragraph. The categories represent how the ~3,000 samples are selected: (1) random, (2) based on size, and (3) large datasets with a high percentage of viral content.

Line 279: Why does it only represent half the raw reads?

Response: Sequence assembly of short reads regularly does not generate contigs that contain all of the reads in the dataset. This has to due with the complexity of the sample and the coverage depth of sequencing applied to the sample. Additional language has been added to clarify why this might occur.

Line 294: Did all of them show similarity to bacteriophage or some were similar to other viruses?

Response: As illustrated in the accompanying figure, the refseqs with the most hits to the contigs were all bacteriophage. This is expected as the data-set used was selected with the goal of enriching for bacteriophage containing data-sets, as described in the methods section.

Lines 310-312 and 319-321: This sentence is there twice

Response: We thank the reviewer for spotting this duplication. We removed the first occurence from the paragraph.

Lines 331 - 348: Was one contig then selected to represent each cluster?

Response: We agree with the reviewer that this approach was not clearly described.  We clustered sequences to create smaller datasets to facilitate downstream analysis steps. To avoid confusion we removed reference to MMSeq as it is not relevant to the results presented (as they were based on MCL not MMSeq). Further, the language was amended to clarify that whole clusters were analyzed, not representatives from clusters.

Line 346: I’m not sure if this is an issue with BLAST, but rather with clustering a large number of data points - a clustering step would be needed regardless of the algorithm used for the pairwise comparisons.

Response: We thank the reviewer to spot this. We amended the language to clarify this point. 

Line 400: Which version of BLAST was used which allows parallelisation?

Response: We considered the use of several cores and threads within  one computational node as parallelization, not across several computational nodes. We adjusted the sentence to explain our understanding of parallelization, in this case as follows:

“While BLAST and DIAMOND can be parallelized to certain degree, i.e. using all cores/threads from a single computing node but not across several computing nodes, HMMER is lacking an efficient multithreaded parallelization.”

Line 412: “virus hunting tool kit” needs a reference

Response: Virus hunting tool kit is self referential to this manuscript. Reference to the “hunting tool kit” part of the sentence has been removed.

Line 453: What is the difference between “NA” and “other”?  If NA means unannotated it is surprising that these are so clearly distinct from the “other” category.

Response: The "center_project_name" field of biosample, where those terms came from, does not use a controlled vocabulary, so submitters may use whatever terms they like. So we agree that it is somewhat surprising that they cluster separately, and have noted this in the discussion.

Line 454: How could this technique be used to correct mislabeled metadata or replace missing metadata when it requires the metadata to classify the sample?

Response: Because of the discrete clustering of samples coming from different sources these correctly annotated datasets can serve as a training set for datasets lacking metadata. Additional details were added to the manuscript to clarify this point.

Line 495: This is not very clear - does it mean most of the hits against RefSeq were good quality hits?

Response: This sentence is not about the quality of the hits. The statement is meant to illustrate that most of the contigs with at least 1 hit to the RefSeq virus database could subsequently be assigned to previously identified viral groups. The language of this sentence has been modified to clarify this.

Figure 1

The use of team names in this figure make it quite confusing, for example the “phylogeny team” and “visualisation team” are not mentioned elsewhere in the text. It is not clear what each step in the pipeline aims to achieve. It would be helpful to specify the role of each team as well as the name.

Response: We believe we have addressed this concern by updating the figure legend for Figure 1. It summarizes the goals of each team at the hackathon and links that summary back to specific sections within the methods and results.

Figure 2

The labels on the figure are too small to see in this version of the document.  Several of the colours in pie chart C are reused so it is not clear which is which - a bar chart might show this data more clearly.

Response: Figures 2 and 6 were reworked to improve legibility. In general we are concerned that the reviewers did not receive the high quality version of our figures.

Figure 4

I’m not sure what is shown in 4B - counts relative to what?  If each contig was classified into one category what are the distributions here?

Response: The contigs were classified based on the absolute number of domain hits (More than 3 eukaryotic domains, or more than 3 prokaryotic domains in combination with no viral domains). The proportions visualised here are the number of domain hits per category divided by the total number of domain hits. This approach was taken to account for the wide distribution of contig lengths (more than 3 orders of magnitude, Figure 4C). We have added a sentence in the figure legend that clarifies this.

Figure 6

The text is too small to see in this version of the document.  

Response: We have rescaled figure 6 and made sure labels and titles have improved readability.

Reviewer 2 Report

This paper presents the report of a virus discovery hackathon. The topic of the hackathon, identifying viruses in public WGS data sets, is timely and the approach is certainly of interest to the scientific community. However, the presentation of the results and the structure of the paper could be substantially improved.

I had issues in identifying the main points of the paper. The main findings could already be mentioned in the abstract. Instead of “valuable insights” (line 53), a summary of the conclusions could be given. The “cloud infrastructure requirements” that were identified should be explicitly mentioned. To this end, the conclusions could be more detailed.

The overview of the teams in Figure 1 is helpful. It can actually appear earlier and be already linked to the Methods sections. Also methods and results should be linked better. I guess that the “Machine Learning” section in the Methods is used in the “Metadata Analysis” section in the Results. Maybe a restructuring with similar titles could help in linking the sections.

The numbers in Figure 1 are useful, but it is not always clear what they refer to (data sets, contigs, genes?). I suggest to present the full numbers that also appear in the text, which would make it easier to follow, how much data is processed in each step. In addition the proper units must be given, e.g., X genes on Y contigs from Z data sets.

The Machine Learning part is not well connected to the rest of the manuscript. In the Methods section, it is unclear what ML is used for.

The section starting on line 340 is especially cryptic. Methods such as MMseqs2 should certainly be used for clustering and it is unclear why they are not reproducible. Which kind of “additional tools” (line 347) are needed? In particular, MMseqs2 is a useful recent development for this task.

The Metadata Analysis is disconnected to the rest of the manuscript. It is not related to identifying viral contigs and some motivation should be given what this section is needed for.

Unclear where the 347,188 “unknown-unknowns” come from (line 368). There are 4M unknown contigs (line 305).

Why is genetic code 12 (Alternative Yeast) used? 11 (bacteria) would be more appropriate.

I could not find Table 1 in the manuscript.

The quality of the figures must be improved. Especially Figures 2 and 6 are not readable.

The reference numbers are not correct, e.g., MMseqs2 [27] should be 28.

Supplementary table 1 should be given in a text format.

Author Response

Overall 

We rearranged and clarified the manuscript in several places to address the issues raised by both reviewers concerning the clarity and purpose of the manuscript. This includes extending the Abstract, improving and clarifying Figure legends and adjusting references. The quality of the figures was improved. We were surprised about the missing Table 1, which was included in our initial submission.  

Reviewer 2

I had issues in identifying the main points of the paper. The main findings could already be mentioned in the abstract. Instead of “valuable insights” (line 53), a summary of the conclusions could be given. The “cloud infrastructure requirements” that were identified should be explicitly mentioned. To this end, the conclusions could be more detailed.

Response: We extended the abstract clarify the main points of the manuscript.  We added clearer short summaries to each result section and adjusted several parts of the manuscript  to further clarify the main points of the paper (q.v. first comment of Reviewer 1). We would be happy to extend the conclusion if it was clearer which parts were thought lacking, as it is we believe the conclusions effectively highlight the main findings and note important implications of the work.

The overview of the teams in Figure 1 is helpful. It can actually appear earlier and be already linked to the Methods sections. Also methods and results should be linked better. I guess that the “Machine Learning” section in the Methods is used in the “Metadata Analysis” section in the Results. Maybe a restructuring with similar titles could help in linking the sections.

Response: The figure legend to Figure 1 was reworked to address these concerns. We feel that this illustrates how the results generated by the various methods are related to each other and not a method in and of itself, thus we chose to keep the first reference to it in the results section. That said, if the reviewers feel strongly about this we are open to referencing it in the methods section.

The numbers in Figure 1 are useful, but it is not always clear what they refer to (data sets, contigs, genes?). I suggest to present the full numbers that also appear in the text, which would make it easier to follow, how much data is processed in each step. In addition the proper units must be given, e.g., X genes on Y contigs from Z data sets.

Response: The numbers provided in Figure 1 always reference the number of contigs analyzed at the step. We have made modifications to the figure legend that should allow for a more accurate interpretation of the figure and provide a reference back to the relevant sections in the text. Numbers have been updated to correspond with those elsewhere in the text.

The Machine Learning part is not well connected to the rest of the manuscript. In the Methods section, it is unclear what ML is used for.

Response: We added language to motivate the work presented in the machine learning part of the results section. Namely, that insufficient metadata complicates making sense of the results presented in the preceding sections.

The section starting on line 340 is especially cryptic. Methods such as MMseqs2 should certainly be used for clustering and it is unclear why they are not reproducible. Which kind of “additional tools” (line 347) are needed? In particular, MMseqs2 is a useful recent development for this task.

Response: We agree with the reviewer  that this section is unclear. The issue with the reproducibility related to the pre-Hackathon, were different methods for filtering etc. were tested using a smaller data set. This resulted in a different data set than during the actual Hackathon. 

The Metadata Analysis is disconnected to the rest of the manuscript. It is not related to identifying viral contigs and some motivation should be given what this section is needed for.

Response: Explanation for the motivation of the work presented in this section was added to the start of the section

Unclear where the 347,188 “unknown-unknowns” come from (line 368). There are 4M unknown contigs (line 305).

Response: We are thankful  to the reviewer to spot this discrepancy. While we attempted to obtain domain alignment results for the complete unknown dataset (4.2 million contigs), with the limited time and compute resources we had available we were able to only obtain results for 347,188 contigs. We have clarified this in the result section. 

Why is genetic code 12 (Alternative Yeast) used? 11 (bacteria) would be more appropriate.

Response: After review of the work presented in this section by a member of the team that conducted the work it was concluded that this was a typo and is now fixed.

I could not find Table 1 in the manuscript.

Response: This is included in the manuscript as far as we can tell; without further clarification on how the document was transmitted to the reviewers we are unclear on how to address this, assuming any changes are actually necessary on our part.

The quality of the figures must be improved. Especially Figures 2 and 6 are not readable.

Response: Figures 2 and 6 we reworked to make labels more legible. In general we are concerned that reviewers did not receive the high quality versions of the figures.

The reference numbers are not correct, e.g., MMseqs2 [27] should be 28.

Response: We thnak the reviewer for spotting this. This was corrected. The references were updated due to the incorporated changes.

Supplementary table 1 should be given in a text format.

Response: This change was made.

Round 2

Reviewer 2 Report

The authors present an improved version of the manuscript. Few small issues remain.

I am glad to read that the genetic code 12 was a typo, however, it is still not fixed in the text (line 922). In addition, the legend of Figure 5 is repeated 3 times in my version of the manuscript.

The countries of the participants are not given in Table 1, although it is mentioned in the text. I suggest to remove the countries from the text or add them to the table.

The authors might want to check the color key given in Fig. 6. Is -0.15 really in the middle and not 0?

Author Response

Hello, and thanks again for further improving this manuscript!

In the new version:

Genetic code has been changed to 11

'countries' has been excluded from the text.

We checked the R script used to generate Fig. 6.  -.15 is really in the middle.  If you would like to take a look, that script is here: https://github.com/NCBI-Hackathons/VirusDiscoveryProject/blob/master/MachineLearning/code/PLS.R

I have run a spelling and grammar check via google docs.  

I have uploaded the revised manuscript.

Thanks again!

Ben and collaborators